# Identification of Tumor Suppressive Genes Regulated by *miR-31-5p* and *miR-31-3p* in Head and Neck Squamous Cell Carcinoma

**DOI:** 10.3390/ijms22126199

**Published:** 2021-06-08

**Authors:** Sachi Oshima, Shunichi Asai, Naohiko Seki, Chikashi Minemura, Takashi Kinoshita, Yusuke Goto, Naoko Kikkawa, Shogo Moriya, Atsushi Kasamatsu, Toyoyuki Hanazawa, Katsuhiro Uzawa

**Affiliations:** 1Department of Oral Science, Graduate School of Medicine, Chiba University, Chiba 260-8670, Japan; Sachi.o8952@chiba-u.jp (S.O.); minemura@chiba-u.jp (C.M.); kasamatsua@faculty.chiba-u.jp (A.K.); uzawak@faculty.chiba-u.jp (K.U.); 2Department of Functional Genomics, Chiba University Graduate School of Medicine, Chiba 260-8670, Japan; cada5015@chiba-u.jp (S.A.); t.kinoshita@chiba-u.jp (T.K.); caxa1117@chiba-u.jp (Y.G.); naoko-k@hospital.chiba-u.jp (N.K.); 3Department of Otorhinolaryngology/Head and Neck Surgery, Chiba University Graduate School of Medicine, Chiba 260-8670, Japan; thanazawa@faculty.chiba-u.jp; 4Department of Biochemistry and Genetics, Chiba University Graduate School of Medicine, Chiba 260-8670, Japan; moriya.shogo@chiba-u.jp

**Keywords:** HNSCC, *miR-31-5p*, *miR-31-3p*, microRNA, oncogenic miRNA, tumor suppressor

## Abstract

We identified the microRNA (miRNA) expression signature of head and neck squamous cell carcinoma (HNSCC) tissues by RNA sequencing, in which 168 miRNAs were significantly upregulated, including both strands of the *miR-31* duplex (*miR-31-5p* and *miR-31-3p*). The aims of this study were to identify networks of tumor suppressor genes regulated by *miR-31-5p* and *miR-31-3p* in HNSCC cells. Our functional assays showed that inhibition of *miR-31-5p* and *miR-31-3p* attenuated cancer cell malignant phenotypes (cell proliferation, migration, and invasion), suggesting that they had oncogenic potential in HNSCC cells. Our in silico analysis revealed 146 genes regulated by *miR-31* in HNSCC cells. Among these targets, the low expression of seven genes (*miR-31-5p* targets: *CACNB2* and *IL34*; *miR-31-3p* targets: *CGNL1, CNTN3, GAS7, HOPX*, and *PBX1*) was closely associated with poor prognosis in HNSCC. According to multivariate Cox regression analyses, the expression levels of five of those genes (*CACNB2*: *p* = 0.0189; *IL34*: *p* = 0.0425; *CGNL1*: *p* = 0.0014; *CNTN3*: *p* = 0.0304; and *GAS7*: *p* = 0.0412) were independent prognostic factors in patients with HNSCC. Our miRNA signature and miRNA-based approach will provide new insights into the molecular pathogenesis of HNSCC.

## 1. Introduction

Head and neck squamous cell carcinoma (HNSCC) arises from the oral cavity, larynx, or pharynx and is ranked the sixth most common cancer [1,2]. In 2018, approximately 84,000 cases of HNSCC were newly diagnosed, and more than 43,000 people died of this disease worldwide [3]. Surgery, radiation therapy, and cisplatin-based chemotherapy are the main treatment strategies for head and neck cancers. At the time of the initial diagnosis, most patients have advanced-stage disease and a poor prognosis (5-year survival rate < 60%) due to lymph node metastasis or recurrence [1]. In addition, cancer cells acquire resistance to cisplatin-based treatment, and the prognosis of patients who fail treatment is extremely poor [4]. The therapeutic effects of molecular-targeted drugs and immune checkpoint inhibitors in patients after treatment failure are poorly understood [5,6].

The Human Genome Project revealed that an extremely large number of non-coding RNAs (ncRNAs) are transcribed from the human genome, and these ncRNAs function in both normal and diseased cells [7,8]. Among ncRNAs, microRNAs (miRNAs) are endogenous single-stranded RNA molecules 19–22 nucleotides long that function as fine tuners of RNA expression in a sequence-dependent manner [9,10]. A unique feature of miRNAs is that a single miRNA negatively regulates a vast number of RNA transcripts (both protein-coding RNAs and ncRNAs) in each cell [10]. Moreover, bioinformatic studies have shown that more than half of protein-coding genes are controlled by miRNAs [11]. Numerous studies have indicated that aberrantly expressed miRNAs disrupt the tightly controlled RNA networks in normal cells, and these events trigger transformation to a disease state [12,13].

Identification of differentially expressed miRNAs in the cancer tissues of interest is the initial step. The latest RNA sequencing technology has successfully resulted in identification of miRNA expression signatures in cancer tissues. Our HNSCC miRNA signature revealed that both strands of the *miR-31* duplex (*miR-31-5p* and *miR-31-3p*) are upregulated in cancer tissues. Numerous cohort data from The Cancer Genome Atlas (TCGA) confirmed that *miR-31-5p* and *miR-31-3p* are upregulated in HNSCC tissues. The aim of this study was to investigate the oncogenic roles of these miRNA strands and to identify their tumor suppressor gene targets in HNSCC cells.

Identification of differentially expressed miRNAs and their regulated molecular networks may be an effective strategy for elucidating the molecular pathogenesis of HNSCC.

## 2. Results

### 2.1. Identification of the miRNA Expression Signature of HNSCC by RNA Sequencing

Six cDNA libraries (derived from three HNSCC tissues and three normal oral epithelial tissues) were analyzed by RNA sequencing. After a trimming procedure, 955,347–1,927,436 reads were successfully mapped to the human miRNAs (Appendix A). The clinical features of the HNSCC specimens using in this study are summarized in Appendix A.

A total of 168 miRNAs were identified as upregulated (log2 fold change > 1.5) in HNSCC tissues (Figure 1A and Appendix A).

### 2.2. Expression Levels and Clinical Significance of miR-31-5p and miR-31-3p in HNSCC

We focused on miRNAs of which both strands (the guide strand and passenger strand) derived from pre-miRNAs were upregulated in this signature. A total of 7 pre-miRNAs (*miR-31, miR-223, miR-4655, miR-4781, miR-6753, miR-6830, and miR-6871*) were detected in this signature (Figure 1A and Appendix A). From TCGA-HNSC database analysis, it was confirmed that *miR-31* is the only miRNA whose expression of both strands were significantly upregulated in HNSCC tissues among 7 pre-miRNAs (Figure 1B). The expression of neither miRNA was associated with worse overall survival rates in patients with HNSCC according to analysis of TCGA-HNSC data (Figure 1C). 

In this study, we focused on *miR-31-5p* and *miR-31-3p,* and continued to investigate the functional aspects of these miRNAs.

### 2.3. Effects of Inhibition of miR-31-5p and miR-31-3p Expression on the Proliferation, Migration, and Invasion of HNSCC Cells

First, we measured the expression levels of *miR-31-5p* and *miR-31-3p* in 11 HNSCC cell lines compared with fibroblast cell lines (IMR-90 and MRC-5). Detailed information on the cell lines used is shown in the Appendix A. Overexpression of *miR-31-5p* and *miR-31-3p* was detected in several HNSCC cell lines, e.g., Ca9-22, HSC-2, HSC-4, and SAS (Appendix A), relative to fibroblasts. We selected two of these HNSCC cell lines, SAS and HSC-2, to investigate the oncogenic roles of these miRNAs. To suppress the expression of *miR-31-5p* and *miR-31-3p,* we used inhibitors (Anti-miR^TM^ miRNA Inhibitor) of these miRNAs. The inhibitors were used at a concentration of 30 nM. To evaluate their effects in functional assays, we confirmed the expression of *miR-31-5p* and *miR-31-3p* after transfection of inhibitors into SAS and HSC-2 cells (Appendix A). Inhibition of *miR-31-5p* and *miR-31-3p* attenuated the proliferation (Figure 2A and Appendix A) and markedly decreased the migration and invasion (Figure 2B,C and Appendix A) of SAS and HSC-2 cells. These data suggest that upregulation of *miR-31-5p and miR-31-3*p has an oncogenic effect in HNSCC cells.

### 2.4. Screening of miR-31-5p and miR-31-3p Targets in HNSCC Cells

Based on our hypothesis that *miR-31-5p* and *miR-31-3p* regulate tumor suppressor genes in HNSCC cells, we screened *miR-31-5p* and *miR-31-3p* target genes using in silico analyses and our gene expression data (GEO accession no. GSE172120). Our strategy for identifying *miR-31-5p/miR-31-3p* gene targets is shown in Figure 3.

Analysis of the TargetScan database revealed that 477 genes and 2387 genes had putative *miR-31-5p* and *miR-31-3p* binding sites, respectively, within their 3′-UTR [14]. Next, we compared these genes with those downregulated in HNSCC clinical tissues, and 146 genes were shared between the data sets (24 and 122 genes were *miR-31-5p* and *miR-31-3p* targets, respectively and they are summarized in Table 1). Furthermore, we performed a clinicopathological analysis of these candidate genes using data from TCGA-HNSC. Seven genes (*CACNB2, IL34, CGNL1, CNTN3, GAS7, HOPX,* and *PBX1*) regulated by *miR-31-5p* and *miR-31-3p* were identified as putative tumor suppressors. Of these genes, five (*CACNB2, IL34, CGNL1, CNTN3,* and *GAS7*) were identified as independent prognostic factors by multivariate analysis.

### 2.5. Clinical Significance of miR-31-5p and miR-31-3p Targets in HNSCC Cells

Among the 146 *miR-31-5p* and *miR-31-3p* gene targets, the low expression of seven (*CACNB2*: *p* = 0.0018; *IL34*: *p* = 0.0031; *CGNL1*: *p* = 0.0012; *CNTN3*: *p* = 0.0061; *GAS7*: *p* = 0.0093; *HOPX*: *p* = 0.0345; and *PBX1*: *p* = 0.0247) significantly predicted a worse prognosis in patients with HNSCC by Kaplan–Meier analysis (Figure 4 and Figure 5). Notably, multivariate Cox regression analyses revealed that the expression levels of five of those genes (*CACNB2*: *p* = 0.0189; *IL34*: *p* = 0.0425; *CGNL1*: *p* = 0.0014; *CNTN3*: *p* = 0.0304; and *GAS7*: *p* = 0.0412) were independent prognostic factors in patients with HNSCC (Figure 6). Moreover, expression negative correlation between *miR-31* and their target genes were investigated by TCGA-HNSC database (Figure 7).

### 2.6. Direct Regulation of CGNL1 by miR-31-3p in HNSCC Cells

We focused on *CGNL1*, which has the most significant difference in clinical statistics, from among the five target genes of *miR-31-5p* and *miR-31-3p**,* and verified the direct regulation of *CGNL1* by *miR-31-3p*. In cells transfected with *miR-**31**-3p*, the levels of *CGNL1* mRNA and CGNL1 protein were significantly lower than in mock- or miR-control-transfected cells (Figure 8A,B).

We performed dual-luciferase reporter assays to determine whether *CGNL1* was directly regulated by *miR-31-3p*. We used vectors encoding the partial wild-type sequences of the 3′-UTR of *CGNL1* and vector with partially deleted *CGNL1* 3′-UTR (Figure 8C). We found that luciferase activity was significantly decreased by cotransfection with *miR-31-3p* and the vector carrying the wild-type 3′-UTR of *CGNL1*, whereas transfection with the deletion vector blocked the decrease in luminescence in SAS cells (Figure 8D). These data showed that *miR-31-3p* directly bound to *CGNL1*.

## 3. Discussion

The latest RNA-sequencing technologies have enabled identification of genome-wide miRNA expression signatures in human cancers. Our recent studies of miRNA signatures revealed that the passenger strands of some miRNA duplexes, such as *miR-99a-3p, miR-145-3p, miR-150-3p* and *miR-199a-3p,* act as tumor suppressors by directly targeting several oncogenes in HNSCC cells [15,16,17,18]. The original theory regarding miRNA biogenesis is that the guide strand of the miRNA duplex is incorporated into the RNA-induced silencing complex and functions as a negative regulator of gene expression, whereas the passenger strand is degraded in the cytoplasm and nonfunctional [10,11]. However, numerous in silico studies (involving over 5200 patients with 14 types of cancers) have shown that both strands (5p and 3p) of some miRNA duplexes (e.g., *miR-30a, miR-139, miR-143,* and *miR-145)* function together to regulate pivotal targets and pathways in several types of cancers [19]. Studies on the passenger strands of miRNAs will reveal novel molecular mechanisms of cancer pathogenesis.

In this study, we focused on *miR-31-5p* and *miR-31-3p* based on our miRNA signatures. Upregulation of *miR-31-5p* and *miR-31-3p* in HNSCC tissues was confirmed by TCGA data analysis. Our functional assays indicated that these miRNAs act as oncogenic miRNAs in HNSCC cells. Previous studies demonstrated that *miR-31* has opposing roles (oncogene vs. tumor suppressor) depending on the type of cancer [20].

In esophageal squamous cell carcinoma, *miR-31* was upregulated in clinical specimens and acted as an oncogenic miRNA by targeting the tumor suppressor gene *LATS2*, which is involved in the Hippo pathway [21]. Upregulation of *miR-31* was reported in HNSCC tissues, and its expression activated the hypoxia-inducible factor pathway by targeting factor-inhibiting hypoxia-inducible factor [22,23]. Signaling via epidermal growth factor and its receptor is an essential oncogenic pathway in HNSCC and oral squamous cell carcinoma (OSCC), and this signaling pathway enhanced AKT activation and upregulated C/EBPβ expression in OSCC [24]. These events induced upregulation of *miR-31* in OSCC cells [24]. Interestingly, a previous study showed that exogenous expression of *miR-31* and telomerase reverse transcriptase transformed normal oral keratinocytes into immortalized cells [25]. Those previous and our present results indicate that upregulation of *miR-31* downregulates genes/pathways intricately involved in malignant transformation of HNSCC and OSCC. 

Our other aim was to clarify the novel molecular pathways regulated by *miR-31-5p* and *miR-31-3p* in HNSCC cells. Our in silico analysis revealed that five genes (*CACNB2, IL34, CGNL1, CNTN3,* and *GAS7*) are closely associated with HNSCC molecular pathogenesis. Functional analyses of these genes are needed to reveal the molecular mechanisms of HNSCC malignant phenotypes.

Of the five genes, *GAS7* was initially cloned from serum-starved mouse NIH3T3 cells, and it consists of a series of different functional domains from the N- to C-termini: Src homology 3 domain, WW domain, and FES-CIP4 homology domain [26]. *GAS7* regulates the dynamic activities of the membrane, actin cytoskeleton, and microtubules [26,27]. Downregulation of *GAS7* has been reported in several cancer types, and ectopic expression of *GAS7* inhibited the migration of lung and breast cancer cells [28]. More recently, it was reported that loss of *GAS7* expression accelerated metastasis of neuroblastoma harboring *MYCN* overexpression or amplification [29]. Previous studies indicated that *GAS7* acts as a tumor suppressor in human cancers.

Analysis of TCGA data showed that *IL34* is downregulated in HNSCC tissues, and its low expression significantly predicts a poor prognosis in patients with HNSCC. *IL34* stimulates the differentiation of monocytes into macrophages via the CSF-1 receptor [30]. *IL34* is also a ligand of the macrophage colony stimulating factor receptor [31]. Recent studies showed that *IL34* is expressed in various types of cancers and is involved in cancer progression and metastasis [32]. In the future, it is necessary to investigate the functional significance of *IL34* in HNSCC.

*CGNL1* is a paralogue of cingulin, which is ubiquitously expressed in endothelial cells and localized at tight junctions [33,34]. A previous study showed that cingulin binds to actin filament bundles to bridge tight junctions and actin filaments [35]. *CGNL1* is localized on actin filament bundles and has multiple roles depending on its binding partner [35]. Previous reports showed that *CGNL1* is an inhibitor of RhoA activity in tight junctions but is also involved in Rac1 activation in Madin–Darby canine kidney epithelial cells [36,37]. *CGNL1* likely has various functions by interacting with different types of GEFs and GAPs in each cell. Few detailed functional analyses of *CGNL1* have been performed in cancer cells. Expression of *CGNL1* was downregulated in HNSCC tissues compared with normal epithelial tissues in numerous TCGA datasets. GEPIA2 database (http://gepia2.cancer-pku.cn/#index, accessed on 20 April 2021) analyses showed that expression of *CGNL1* was significantly downregulated in cervical squamous cell carcinoma, esophageal carcinoma, and lung squamous cell carcinoma, suggesting that *CGNL1* is downregulated in human squamous cell carcinoma. These findings suggest that *CGNL1* plays a tumor suppressor role in HNSCC cells [38]. Sufficient functional analysis of *CGNL1* in HNSCC remains unresolved in this study. By clarifying the tumor suppressive function of this gene in the future, a part of the molecular mechanism of HNSCC will be clarified.

We newly created the miRNA expression signature of HNSCC by RNA sequencing. Analysis of the signature revealed that both strands of pre-*miR-31* (the guide strand of *miR-31-5p* and the passenger strand of *miR-31-3p*) acted as oncogenic miRNAs in HNSCC cells. Our in silico analysis showed that a total of 5 genes (*CACNB2, IL34, CGNL1, CNTN3,* and *GAS7*) were independent prognostic factors in patients with HNSCC. Our HNSCC miRNA signature and miRNA-based analyses will provide important insights into the molecular pathogenesis of HNSCC.

## 4. Materials and Methods

### 4.1. Clinical HNSCC and Normal Epithelial Tissue Specimens and HNSCC Cell Lines

Six specimens (three HNSCC tissues and three normal oral epithelial tissues) were analyzed by RNA sequencing to determine the HNSCC miRNA signature. The clinical features of HNSCC patients are summarized in Appendix A.

All specimens used were obtained by surgical resection at Chiba University Hospital. All patients provided written informed consent for the use of their specimens. This study was approved by the Bioethics Committee of Chiba University (approval number: 28–65, 10 February 2015).

Two human HNSCC cell lines (HSC-2 and SAS) were obtained from the RIKEN BioResource Center (Tsukuba, Ibaraki, Japan) and used in this study.

### 4.2. Determination of the miRNA Expression Signature in HNSCC by RNA Sequencing

Small RNAs were sequenced to determine the miRNA expression signature of HNSCC. The RNA sequencing procedure was described in our previous studies [39,40,41,42].

### 4.3. RNA Extraction and Quantitative Reverse-Transcription Polymerase Chain Reaction (qRT-PCR)

RNA was extracted from clinical specimens and cell lines [15,16,17,18] and subjected to qRT-PCR for miRNA expression analysis [15,16,17,18] as described previously. The TaqMan probes, primers used in this study are listed in Appendix A.

### 4.4. Transfection of Mirnas Precursors and Inhibitors into HNSCC Cells

The procedures used for transfection of miRNA precursors and inhibitors into HNSCC cells have been described previously [15,16,17,18]. The reagents used in this study are listed in Appendix A.

### 4.5. Functional Assays (Cell Proliferation, Migration, and Invasion) in HNSCC Cells

The procedures used for the functional assays in cancer cells (proliferation, migration, and invasion) have been described in our previous studies [15,16,17,18]. Cells were transfected with 30 nM miRNA inhibitors. Cell proliferation was evaluated by XTT assay. Migration assays were performed using uncoated transwell polycarbonate membrane filters, and invasion assays were conducted using modified Boyden chambers.

### 4.6. Analysis of the Clinical Significance of HNSCC Patients Using TCGA-HNSC Data

The strategy used to identify miRNA target genes is presented in Figure 3. We selected putative target genes with *miR-31-5p* and *miR-31-3p* binding sites using TargetScanHuman ver. 7.2 (http://www.targetscan.org/vert_72/; data were downloaded on 10 July 2020). The expression profiles of HNSCC clinical specimens (genes downregulated in HNSCC tissues) were used for screening miRNA target genes [14]. Our expression data were deposited in the GEO database (accession number: GSE172120). Furthermore, we narrowed down the candidate genes by factoring in clinical information from TCGA-HNSC analyses.

For the Kaplan–Meier survival analysis, we downloaded TCGA-HNSC clinical data (TCGA, Firehose Legacy) from cBioportal (https://www.cbioportal.org, accessed on 10 April 2020). Gene expression data for each gene were collected from OncoLnc (http://www.oncolnc.org, accessed on 20 April 2021) [43]. For the log-rank test, we used R ver. 4.0.2 (R Foundation for Statistical Computing, Vienna, Austria), and “survival” and “survminer” packages.

Multivariate Cox regression analyses were also performed using TCGA-HNSC clinical data and survival data according to the expression level of each gene from OncoLnc to identify factors associated with HNSCC patient survival [43]. In addition to gene expression, the tumor stage, pathological grade, and age were evaluated as potential independent prognostic factors. The multivariate analyses were performed using JMP Pro 15.0.0 (SAS Institute Inc., Cary, NC, USA).

### 4.7. Western Blotting

Cell lysates were prepared 48 h after transfection with RIPA buffer (Nacalai Tesque, Chukyo-ku, Kyoto, Japan). Then, 20 μg of protein lysates were separated on 4–12% Bis-Tris gel and transferred to nitrocellulose membranes (Invitrogen, Carlsbad, CA, USA) and blocked for 1 h at room temperature with Blocking One (Nacalai Tesque, Inc., Kyoto, Japan). The antibodies used in this study are shown in Appendix A.

### 4.8. Plasmid Construction and Dual-Luciferase Reporter Assays

The partial wild-type sequence of the *CGNL1* 3′-untranslated region (3′-UTR) was inserted between the XhoI-PmeI restriction sites in the 3′-UTR of the hRluc gene in the psiCHECK-2 vector (C8021; Promega, Madison, WI, USA). Alternatively, we used sequences that were missing the *miR-31-3p* target sites. The synthesized DNA was cloned into the psiCHECK-2 vector. SAS cells were transfected with 50 ng of the vector, 10 nM microRNAs, and 0.5 µL Lipofectamine 2000 in 50 µL Opti-MEM (both from Invitrogen, Carlsbad, CA, USA).

### 4.9. Statistical Analysis

Statistical analyses were performed using GraphPad Prism 7 (GraphPad Software, La Jolla, CA, USA) and JMP Pro 15 (SAS Institute Inc., Cary, NC, USA). Dunnet’s test were used for multiple group comparisons. For correlation analyses, Spearman’s test was applied. A *p* value less than 0.05 was considered statistically significant. Bar graphs (Figure 2, Figure 8A,D, Appendix A and Appendix A) showed mean value and standard error.

## 5. Conclusions

In this study, we focused on *miR-31-5p* and *miR-31-3p* based on our miRNA signatures. Our functional assays indicated that these miRNAs play an oncogenic role in HNSCC cells. Using in silico database analysis to identify gene targets regulated by *miR-31-5p and miR-31-3p,* we rapidly identified candidate tumor suppressor genes in HNSCC. Our HNSCC miRNA signature and miRNA-based analyses will provide important insights into the molecular pathogenesis of HNSCC.

## Figures and Tables

**Figure 1 ijms-22-06199-f001:**
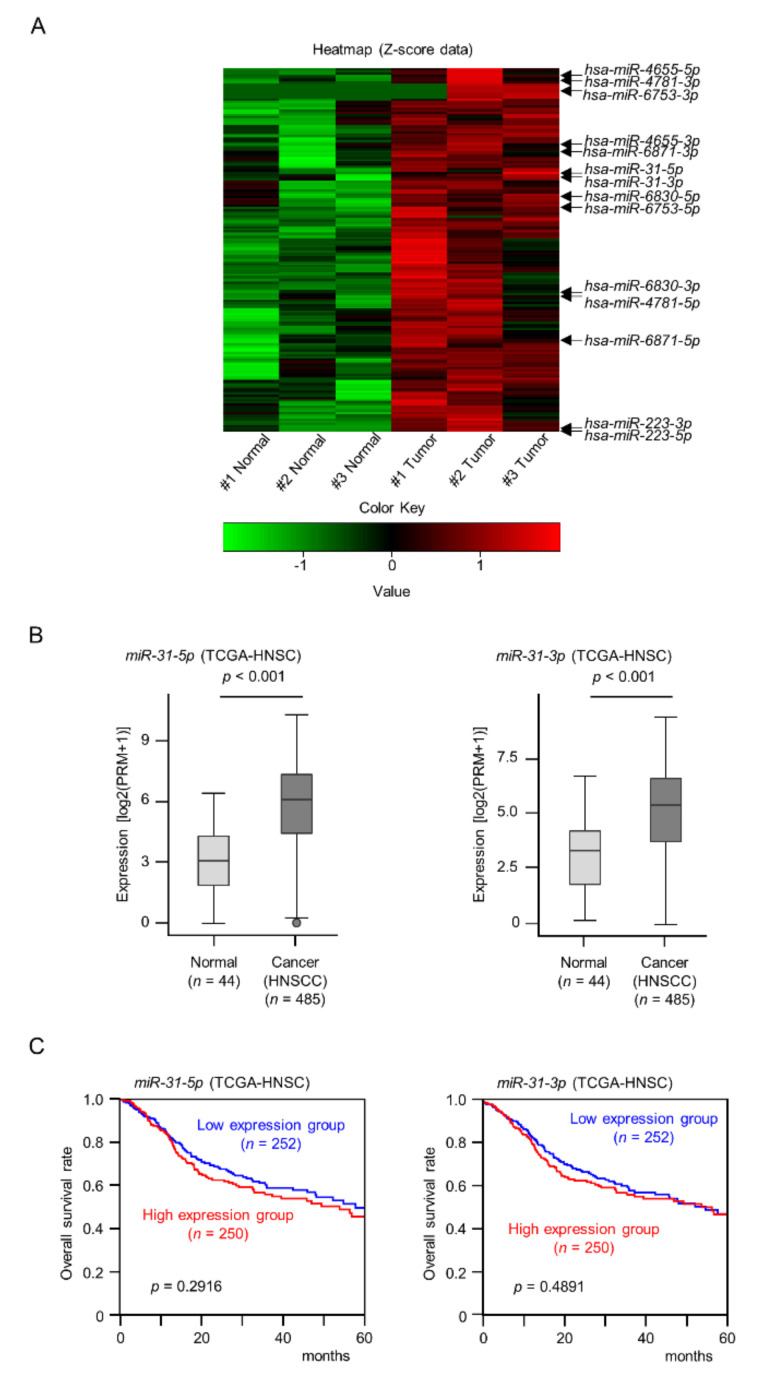
Clinical significance of *miR-31-5p* and *miR-31-3p* expression in HNSCC clinical specimens. (**A**) Heat maps of the 168 upregulated miRNAs in HNSCC clinical specimens. The color scale was based on Z-score of miRNA-seq expression data. (**B**) Expression levels of *miR-31-5p* and *miR-31-3p* were evaluated using TCGA-HNSC data. (**C**) Kaplan–Meier survival analyses of HNSCC patients using data from TCGA-HNSC. Patients were divided into two groups according to the median miRNA expression level: high and low expression groups. The red and blue lines represent the high and low expression groups, respectively.

**Figure 2 ijms-22-06199-f002:**
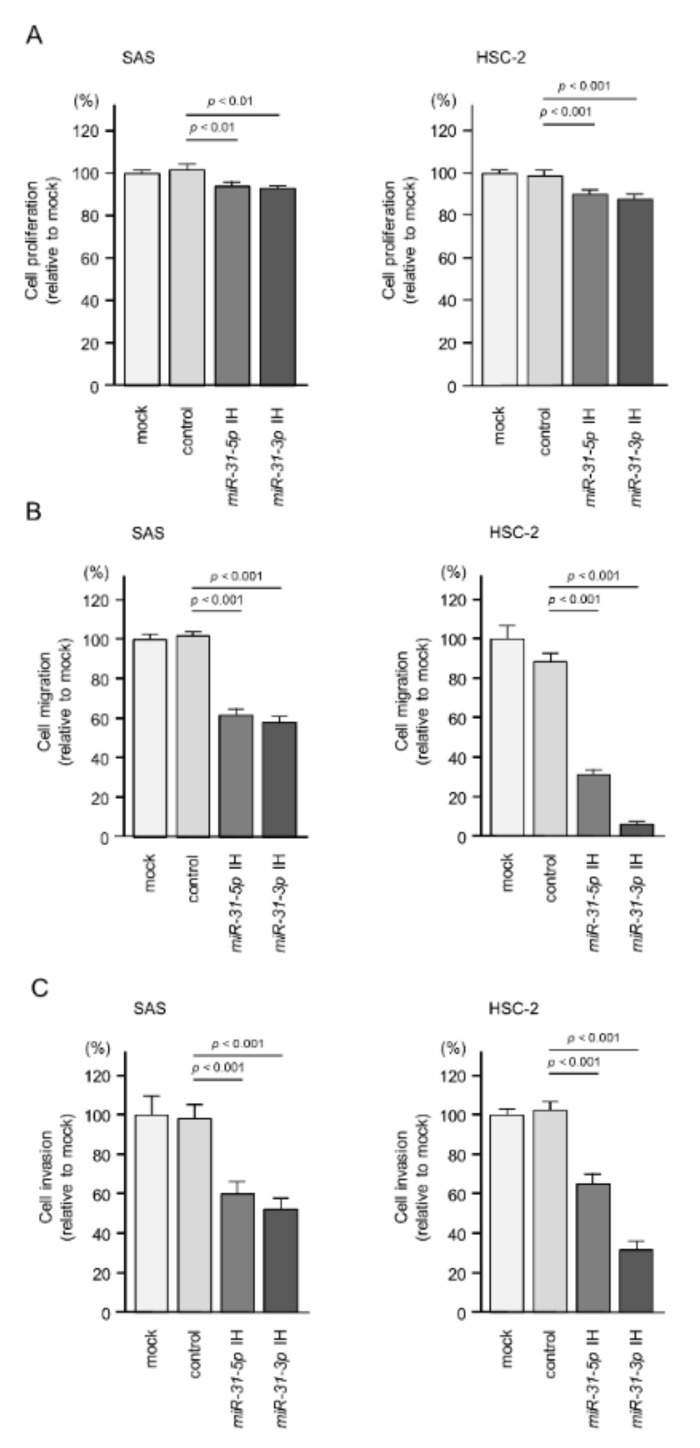
Functional assays of *miR-31-5p* and *miR-31-3p* in HNSCC cell lines (SAS and HSC-2). (**A**) Cell proliferation was assessed using XTT assays at 72 h after the inhibitor transfection. (**B**) Cell migration was assessed using a membrane culture system at 48 h after seeding the inhibitor-transfected cells into the chambers. (**C**) Cell invasion was determined using Matrigel invasion assays at 48 h after seeding the inhibitor-transfected cells into the chambers.

**Figure 3 ijms-22-06199-f003:**
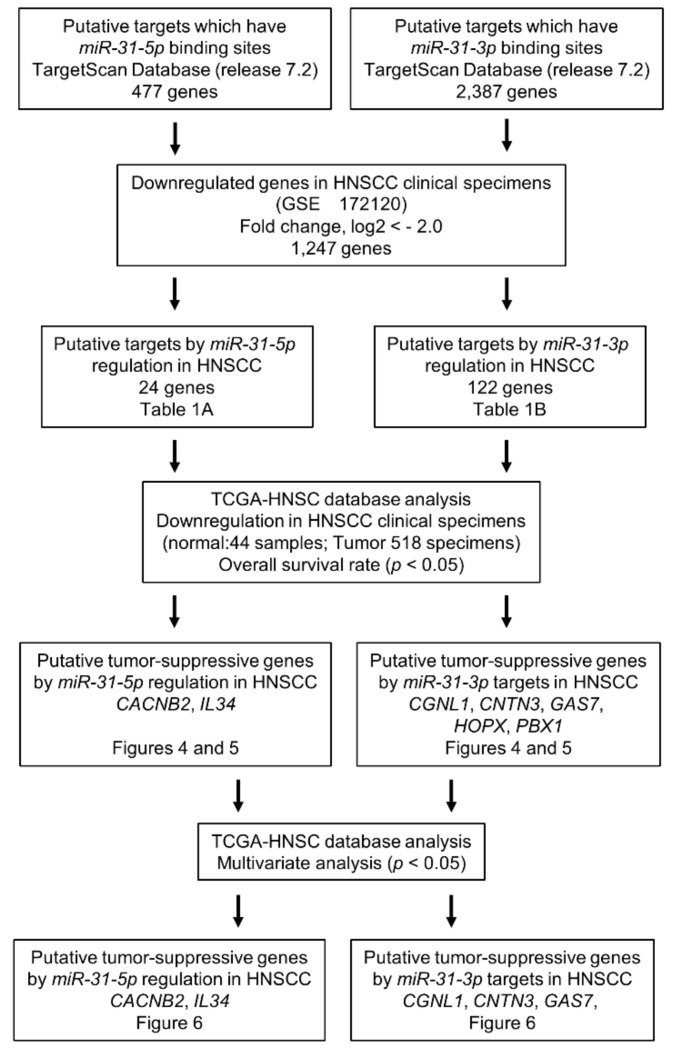
Flow chart of the strategy used to identify putative tumor suppressor genes regulated by *miR-31-5p* and *miR-31-3p* in HNSCC cells.

**Figure 4 ijms-22-06199-f004:**
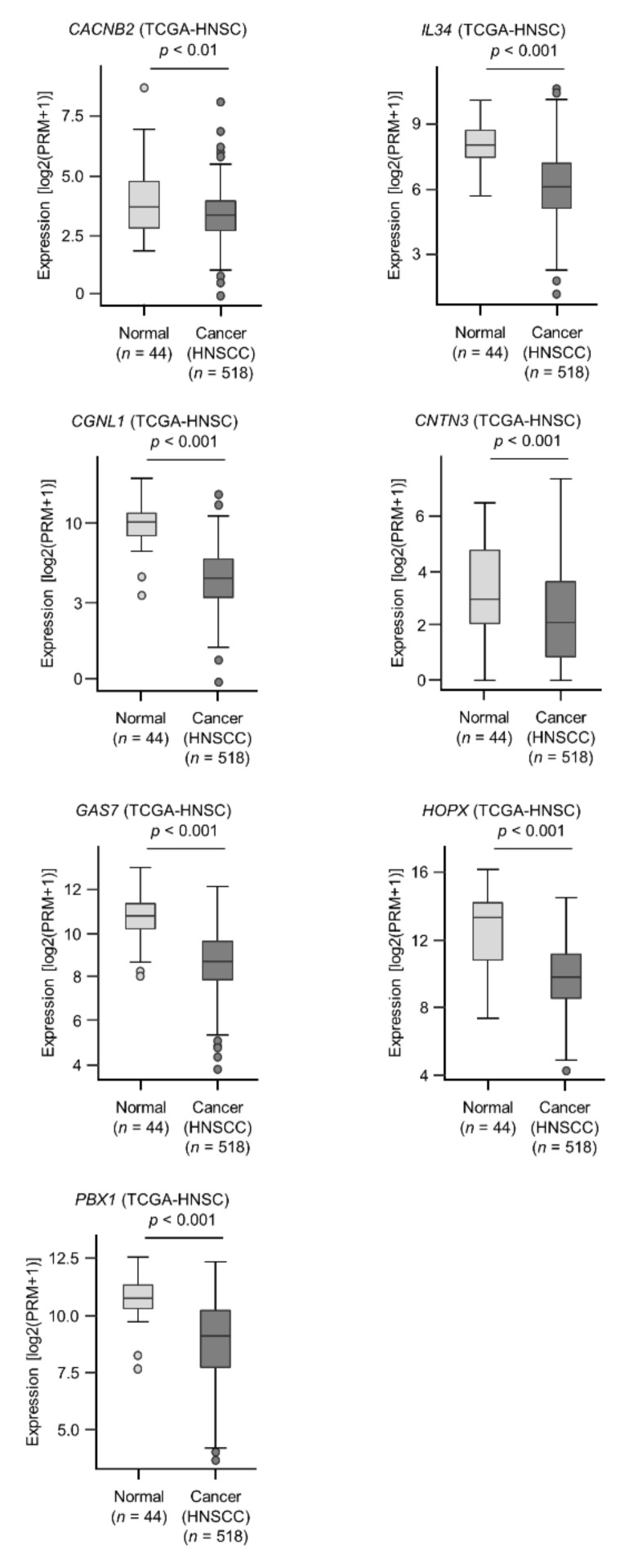
Expression levels of seven target genes (*CACNB2, IL34, CGNL1, CNTN3, GAS7, HOPX,* and *PBX1*) in HNSCC clinical specimens from TCGA-HNSC. All genes were found to be downregulated in HNSCC tissues (*n* = 518) compared with normal tissues (*n* = 44).

**Figure 5 ijms-22-06199-f005:**
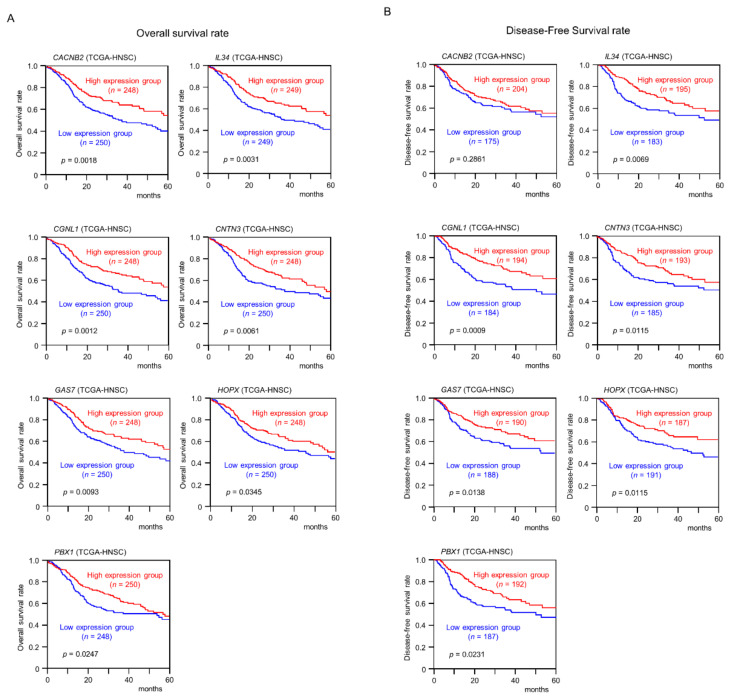
Clinical significance of seven target genes (*CACNB2, IL34, CGNL1, CNTN3, GAS7, HOPX,* and *PBX1*) according to TCGA-HNSC data analysis. (**A**) Kaplan–Meier curves of the 5-year overall survival rate according to the expression of each gene are presented. Low expression of all seven genes was significantly predictive of a worse prognosis in patients with HNSCC. Patients were divided into two groups according to the median miRNA expression level: high and low expression groups. The red and blue lines represent the high and low expression groups, respectively. (**B**) Kaplan–Meier curves of the 5-year disease free survival rate according to the expression of each gene are presented. Low expression of six genes other than *CACNB2* was significantly predictive of a worse prognosis in patients with HNSCC.

**Figure 6 ijms-22-06199-f006:**
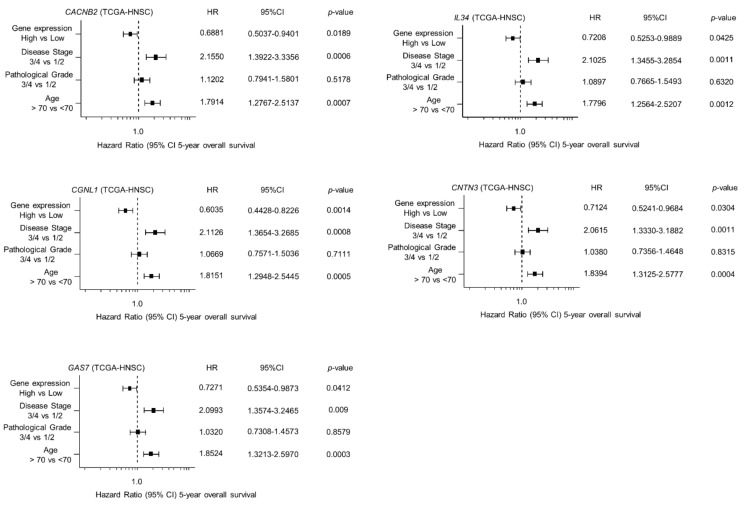
Forest plot showing the multivariate analysis results for the five target genes (*CACNB2, IL34, CGNL1, CNTN3*, and *GAS7*) identified by analysis of TCGA-HNSC data. The multivariate analysis determined that the expression levels of five genes were independent prognostic factors in terms of the 5-year overall survival rate after the adjustment for tumor stage, age, and pathological stage (*p* < 0.05).

**Figure 7 ijms-22-06199-f007:**
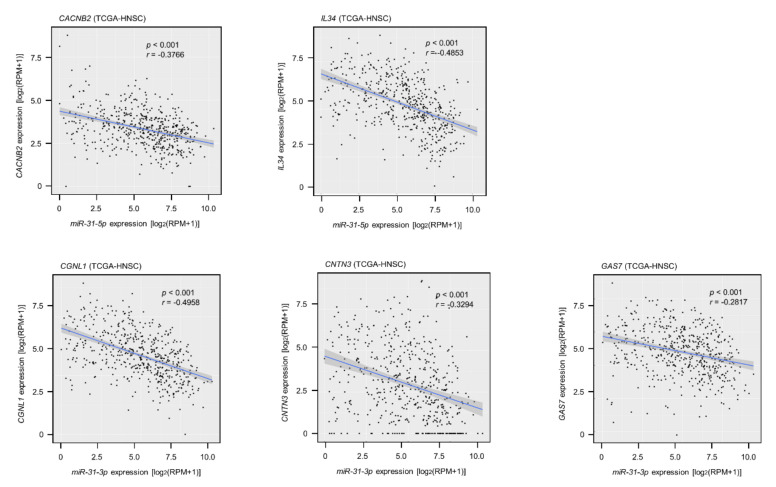
Expression correlation between *miR-31* and their target genes in HNSCC clinical specimens. Spearman’s rank test indicated negative correlations of *miR-31-5p* expression with their targets (*CACNB2*/*miR-31-5p*: *p* < 0.001, *r* = −0.3748; *IL34*/*miR-31-5p*: *p* < 0.001, *r* = −0.5296). Similarly, negative correlations were detected in *miR-31-3p* expression with their targets (*CGNL1*/*miR-31-3p*: *p* < 0.001, *r* = −0.5145; *CNTN3/miR-31-3p*: *p* < 0.001, *r* = −0.3601; *GAS7*/*miR-31-3p*: *p* < 0.001, *r* = −0.3170).

**Figure 8 ijms-22-06199-f008:**
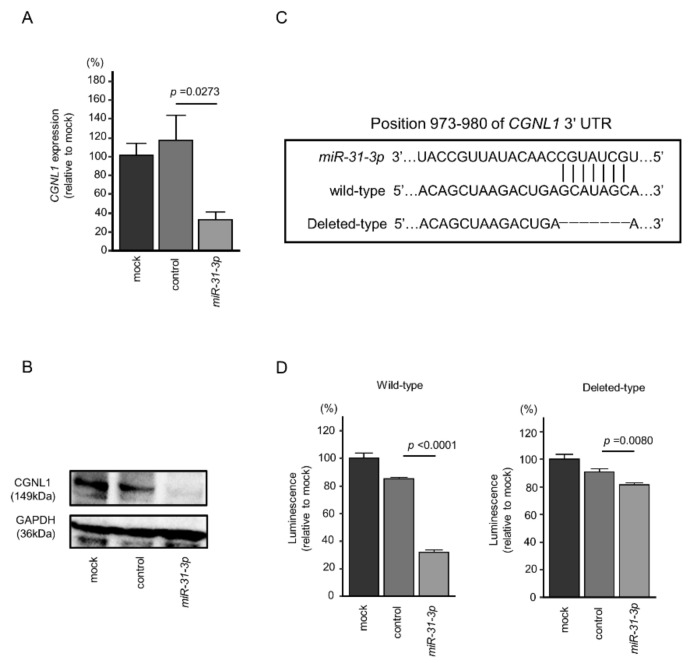
Expression of *CGNL1* was regulated directly by *miR-31-3p* in HNSCC cells. (**A**) Expression of *CGNL1* mRNA was significantly suppressed in *miR-31-3p*-transfected SAS cells (48 h after transfection). (**B**) Expression of CGNL1 protein was reduced in *miR-31-3p*-transfected HNSCC cells (48 h after transfection). GAPDH was used as a loading control. (**C**) The Target Scan Human database predicted one putative *miR-31-3p*-binding site in the 3′-UTR of *CGNL1* [14]. (**D**) Dual-luciferase reporter assays showed decreased luminescence activity in SAS cells co-transfected with *miR-31-3p* together with a vector harboring the “wild-type”. Normalized data were calculated as Renilla/firefly luciferase activity ratios.

**Table 1 ijms-22-06199-t001:** A. Candidate target genes regulated by *miR-31-5p*. B. Candidate target genes regulated by *miR-31-3p*.

A				
Entrez Gene ID	Gene Symbol	Gene Name	Fold Change (log2 < −2.0)	Total Sites
5563	*PRKAA2*	protein kinase, AMP-activated, alpha 2 catalytic subunit	−4.56	1
83699	*SH3BGRL2*	SH3 domain binding glutamate-rich protein like 2	−4.45	1
6517	*SLC2A4*	solute carrier family 2 (facilitated glucose transporter), member 4	−4.29	1
2252	*FGF7*	fibroblast growth factor 7	−3.81	1
55607	*PPP1R9A*	protein phosphatase 1, regulatory subunit 9A	−3.73	1
5549	*PRELP*	proline/arginine-rich end leucine-rich repeat protein	−3.66	2
5083	*PAX9*	paired box 9	−3.62	1
26084	*ARHGEF26*	Rho guanine nucleotide exchange factor (GEF) 26	−3.58	1
252995	*FNDC5*	fibronectin type III domain containing 5	−3.50	1
51209	*RAB9B*	RAB9B, member RAS oncogene family	−3.23	1
2899	*GRIK3*	glutamate receptor, ionotropic, kainate 3	−2.88	1
401474	*SAMD12*	sterile alpha motif domain containing 12	−2.84	1
60529	*ALX4*	ALX homeobox 4	−2.63	1
64399	*HHIP*	hedgehog interacting protein	−2.53	1
146433	*IL34*	interleukin 34	−2.45	1
84144	*SYDE2*	synapse defective 1, Rho GTPase, homolog 2 (C. elegans)	−2.44	2
619279	*ZNF704*	zinc finger protein 704	−2.40	1
783	*CACNB2*	calcium channel, voltage-dependent, beta 2 subunit	−2.34	1
5493	*PPL*	periplakin	−2.27	1
3670	*ISL1*	ISL LIM homeobox 1	−2.24	2
389208	*TMPRSS11F*	transmembrane protease, serine 11F	−2.17	1
168667	*BMPER*	BMP binding endothelial regulator	−2.16	1
1983	*EIF5*	eukaryotic translation initiation factor 5	−2.14	1
5100	*PCDH8*	protocadherin 8	−2.06	2
**B**				
**Entrez** **Gene ID**	**Gene Symbol**	**Gene Name**	**Fold Change (log2 < −2.0)**	**Total** **Sites**
420	*ART4*	ADP-ribosyltransferase 4 (Dombrock blood group)	−6.92	1
5075	*PAX1*	paired box 1	−6.47	1
1805	*DPT*	dermatopontin	−5.69	1
10218	*ANGPTL7*	angiopoietin-like 7	−5.09	1
2315	*MLANA*	melan-A	−4.81	1
55286	*C4orf19*	chromosome 4 open reading frame 19	−4.76	1
8839	*WISP2*	WNT1 inducible signaling pathway protein 2	−4.72	1
440854	*CAPN14*	calpain 14	−4.70	1
6422	*SFRP1*	secreted frizzled-related protein 1	−4.70	1
114905	*C1QTNF7*	C1q and tumor necrosis factor related protein 7	−4.66	1
9068	*ANGPTL1*	angiopoietin-like 1	−4.56	1
5563	*PRKAA2*	protein kinase, AMP-activated, alpha 2 catalytic subunit	−4.56	2
5104	*SERPINA5*	serpin peptidase inhibitor, clade A (alpha-1 antiproteinase, antitrypsin), member 5	−4.45	1
148213	*ZNF681*	zinc finger protein 681	−4.27	1
127435	*PODN*	podocan	−4.20	1
53405	*CLIC5*	chloride intracellular channel 5	−4.16	1
85477	*SCIN*	scinderin	−4.09	1
255798	*SMCO1/C3orf43*	single-pass membrane protein with coiled-coil domains 1	−4.01	1
53353	*LRP1B*	low density lipoprotein receptor-related protein 1B	−4.00	1
23242	*COBL*	cordon-bleu WH2 repeat protein	−3.89	1
5570	*PKIB*	protein kinase (cAMP-dependent, catalytic) inhibitor beta	−3.84	1
440730	*TRIM67*	tripartite motif containing 67	−3.83	1
2252	*FGF7*	fibroblast growth factor 7	−3.81	1
84525	*HOPX*	HOP homeobox	−3.81	1
389432	*SAMD5*	sterile alpha motif domain containing 5	−3.79	1
8736	*MYOM1*	myomesin 1	−3.68	1
5549	*PRELP*	proline/arginine-rich end leucine-rich repeat protein	−3.66	2
137735	*ABRA*	actin-binding Rho activating protein	−3.58	1
785	*CACNB4*	calcium channel, voltage-dependent, beta 4 subunit	−3.57	3
79442	*LRRC2*	leucine rich repeat containing 2	−3.55	2
339512	*C1orf110*	chromosome 1 open reading frame 110	−3.50	1
10894	*LYVE1*	lymphatic vessel endothelial hyaluronan receptor 1	−3.43	1
3768	*KCNJ12*	potassium channel, inwardly rectifying subfamily J, member 12	−3.36	1
171024	*SYNPO2*	synaptopodin 2	−3.35	1
114786	*XKR4*	XK, Kell blood group complex subunit-related family, member 4	−3.33	1
84952	*CGNL1*	cingulin-like 1	−3.30	2
55335	*NIPSNAP3B*	nipsnap homolog 3B (C. elegans)	−3.27	1
3479	*IGF1*	insulin-like growth factor 1 (somatomedin C)	−3.26	2
2690	*GHR*	growth hormone receptor	−3.20	1
8522	*GAS7*	growth arrest-specific 7	−3.18	1
2066	*ERBB4*	erb-b2 receptor tyrosine kinase 4	−3.14	2
202333	*CMYA5*	cardiomyopathy associated 5	−3.13	1
22865	*SLITRK3*	SLIT and NTRK-like family, member 3	−3.13	1
51666	*ASB4*	ankyrin repeat and SOCS box containing 4	−3.08	1
22871	*NLGN1*	neuroligin 1	−3.08	1
4958	*OMD*	osteomodulin	−3.08	1
5178	*PEG3*	paternally expressed 3	−3.06	1
29119	*CTNNA3*	catenin (cadherin-associated protein), alpha 3	−3.04	2
8529	*CYP4F2*	cytochrome P450, family 4, subfamily F, polypeptide 2	−3.01	1
343450	*KCNT2*	potassium channel, sodium activated subfamily T, member 2	−3.00	1
5087	*PBX1*	pre-B-cell leukemia homeobox 1	−2.98	1
387758	*FIBIN*	fin bud initiation factor homolog (zebrafish)	−2.96	1
57689	*LRRC4C*	leucine rich repeat containing 4C	−2.96	1
79071	*ELOVL6*	ELOVL fatty acid elongase 6	−2.95	1
6542	*SLC7A2*	solute carrier family 7 (cationic amino acid transporter, y+ system), member 2	−2.94	1
6450	*SH3BGR*	SH3 domain binding glutamate-rich protein	−2.93	1
7276	*TTR*	transthyretin	−2.92	2
23732	*FRRS1L/C9orf4*	ferric-chelate reductase 1-like	−2.89	1
220963	*SLC16A9*	solute carrier family 16, member 9	−2.88	1
55	*ACPP*	acid phosphatase, prostate	−2.84	1
401474	*SAMD12*	sterile alpha motif domain containing 12	−2.84	1
8153	*RND2*	Rho family GTPase 2	−2.83	1
7135	*TNNI1*	troponin I type 1 (skeletal, slow)	−2.82	1
340596	*LHFPL1*	lipoma HMGIC fusion partner-like 1	−2.77	1
26974	*ZNF285*	zinc finger protein 285	−2.74	1
2053	*EPHX2*	epoxide hydrolase 2, cytoplasmic	−2.73	1
386618	*KCTD4*	potassium channel tetramerization domain containing 4	−2.73	1
1183	*CLCN4*	chloride channel, voltage-sensitive 4	−2.69	1
291	*SLC25A4*	solute carrier family 25 (mitochondrial carrier; adenine nucleotide translocator), member 4	−2.68	1
4023	*LPL*	lipoprotein lipase	−2.65	1
32	*ACACB*	acetyl-CoA carboxylase beta	−2.64	1
55244	*SLC47A1*	solute carrier family 47 (multidrug and toxin extrusion), member 1	−2.64	1
84620	*ST6GAL2*	ST6 beta-galactosamide alpha-2,6-sialyltranferase 2	−2.62	1
26032	*SUSD5*	sushi domain containing 5	−2.61	1
6857	*SYT1*	synaptotagmin I	−2.61	2
6391	*SDHC*	succinate dehydrogenase complex, subunit C, integral membrane protein, 15kDa	−2.60	1
5506	*PPP1R3A*	protein phosphatase 1, regulatory subunit 3A	−2.58	2
367	*AR*	androgen receptor	−2.57	2
64399	*HHIP*	hedgehog interacting protein	−2.53	1
56898	*BDH2*	3-hydroxybutyrate dehydrogenase, type 2	−2.52	2
9077	*DIRAS3*	DIRAS family, GTP-binding RAS-like 3	−2.52	1
154661	*RUNDC3B*	RUN domain containing 3B	−2.52	1
8796	*SCEL*	sciellin	−2.52	1
50937	*CDON*	cell adhesion associated, oncogene regulated	−2.49	1
6660	*SOX5*	SRY (sex determining region Y)-box 5	−2.48	1
56172	*ANKH*	ANKH inorganic pyrophosphate transport regulator	−2.46	1
6092	*ROBO2*	roundabout, axon guidance receptor, homolog 2 (Drosophila)	−2.46	1
158326	*FREM1*	FRAS1 related extracellular matrix 1	−2.45	1
10345	*TRDN*	triadin	−2.45	1
158866	*ZDHHC15*	zinc finger, DHHC-type containing 15	−2.44	1
55283	*MCOLN3*	mucolipin 3	−2.42	1
653316	*FAM153C*	family with sequence similarity 153, member C, pseudogene	−2.41	1
348158	*ACSM2B*	acyl-CoA synthetase medium-chain family member 2B	−2.39	1
11227	*GALNT5*	polypeptide N-acetylgalactosaminyltransferase 5	−2.39	1
3169	*FOXA1*	forkhead box A1	−2.37	1
284716	*RIMKLA*	ribosomal modification protein rimK-like family member A	−2.37	2
253559	*CADM2*	cell adhesion molecule 2	−2.36	1
144453	*BEST3*	bestrophin 3	−2.35	1
2258	*FGF13*	fibroblast growth factor 13	−2.35	1
57863	*CADM3*	cell adhesion molecule 3	−2.34	1
140456	*ASB11*	ankyrin repeat and SOCS box containing 11, E3 ubiquitin protein ligase	−2.32	2
346389	*MACC1*	metastasis associated in colon cancer 1	−2.30	2
9378	*NRXN1*	neurexin 1	−2.30	1
151887	*CCDC80*	coiled-coil domain containing 80	−2.29	2
266977	*GPR110*	G protein-coupled receptor 110	−2.28	1
3481	*IGF2*	insulin-like growth factor 2	−2.27	1
57554	*LRRC7*	leucine rich repeat containing 7	−2.27	1
80310	*PDGFD*	platelet derived growth factor D	−2.25	1
342926	*ZNF677*	zinc finger protein 677	−2.25	1
341640	*FREM2*	FRAS1 related extracellular matrix protein 2	−2.24	1
5067	*CNTN3*	contactin 3 (plasmacytoma associated)	−2.22	1
4919	*ROR1*	receptor tyrosine kinase-like orphan receptor 1	−2.20	1
948	*CD36*	CD36 molecule (thrombospondin receptor)	−2.19	1
23171	*GPD1L*	glycerol-3-phosphate dehydrogenase 1-like	−2.18	1
64102	*TNMD*	tenomodulin	−2.18	2
55638	*SYBU*	syntabulin (syntaxin-interacting)	−2.17	1
6586	*SLIT3*	slit homolog 3 (Drosophila)	−2.13	2
2247	*FGF2*	fibroblast growth factor 2 (basic)	−2.11	1
115827	*RAB3C*	RAB3C, member RAS oncogene family	−2.11	2
203859	*ANO5*	anoctamin 5	−2.10	1
80110	*ZNF614*	zinc finger protein 614	−2.10	1
115265	*DDIT4L*	DNA-damage-inducible transcript 4-like	−2.03	1

## Data Availability

Our expression data were deposited in the GEO database (accession number: GSE172120).

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
