# Peer review of "Identification of Tumor Suppressive Genes Regulated by miR-31-5p and miR-31-3p in Head and Neck Squamous Cell Carcinoma"

_ijms, 2021, doi:10.3390/ijms22126199_

Round 1

Reviewer 1 Report

In the submitted manuscript Oshima et al., by combining bioinformatical analyses of new and publicly available miRNA and mRNA expression data, together with in vitro assays, discovered that putative miR-31-5p and miR-31-3p target genes CACNB2, IL34, CGNL1, CNTN3 and GAS7 have potential tumor suppressive role in head and neck squamous cell carcinoma (HNSCC), and that inhibition of those two miRNAs decrease the proliferation, migration and invasion of HNSCC cells.

Although the presented results are somehow interesting, this study is quite simplistic and the hypothesis lacks more solid biological basis.

I have tree major concerns with this manuscript:

1) I still don't know why the authors have studied particularly those two miRNAs when they discovered 168 miRNAs up-regulated in HNSCC tissue samples, of which miR-31-5p/3p are somewhere in the middle of list according to logFC value! And what about significant down-regulated miRNAs and up-regulated mRNAs?! In the present form this manuscript looks more like a salami publication.

2) Such large tables like Table 1, 2A and 2B are inappropriate to be placed in the main manuscript. It would be better to have heatmaps or PCA plots for distinguishing types of used samples based on the whole transcriptome. Those tables should be provided as supplementary. Also, since authors were interested only in down-regulated genes, there is an error in Table 2A and 2B column name: "(log2> 2.0)"!

3) Authors have potentially used to many self-citations, especially when describing methods (References 14-17), so now potential reader cannot completely and clearly follow what authors have done newly and what are their previous results! When someone looks at their published papers, it could be seen that they have previously used much more HNSCC tissue samples so it is not clear why now they used only 6?! In continuation with my previous comment, this manuscript shows elements of "cherry picking".

To address some other bigger or smaller drawbacks:

Line 21: It is not evident how were those genes "closely associated with HNSCC molecular pathogenesis" when you only showed potential impact on an OVERALL survival (not even a disease-specific)?

Line 31: Please provide the newest Cancer Statistics paper: https://pubmed.ncbi.nlm.nih.gov/33433946/

Line 53 and throughout the text: Please do not use word "signature" when describe or present only 1-2 miRNA/mRNA expressions alone! It would be appropriate if you have shown that combinatorial expression of several miRNAs/mRNAs together have relation to HNSCC!

Line 64: It is not clear why your miRNA expression data is not deposited to GEO database while gene (mRNA) expression is (GSE172120)?! Or this is just not clearly described in the text...

Line 67: Put "features of the HNSCC specimens".

Line 68: Again it is not clear why you focused only on up-regulated miRNAs?!

Line 77 and throughout the text: It is not clear which all or particular TCGA dataset(s) you have used?! Is it only TCGA-HNSC? Please provide a citation for this dataset: https://www.nature.com/articles/nature14129

Also clearly state on figures which TCGA dataset you used for the analysis presented on the figure, not only "HNSCC" because it could be ambiguous.

Line 94: Describe in the text why you chose particularly SAS and HSC-2 HNSCC cell lines because when you look at Figure S1 you can see that they do not have the highest mean expression levels of miR-31-5p and miR-31-3p!

Such large dispersion (standard deviation) for miRNA expresion in a single cell line is strange and unacceptable!

Also, where are the supplementary figures' legends?

Line 96: Although you mostly cite your previous papers when describing methods, some more information are needed in general.

What is "mock" and what "control"?

Regarding the Supplementary Table 3, for what were those Pre-miR miRNA Precursors used?

Lines 97-98: Something is missing in that sentence.

Line 119: Why experimentally proven data about miRNA:mRNA interactions were not used? For instance, data from TarBase v.8 database would provide more plausibility and importance of your selection of genes.

Line 123: You said that you have "performed a clinicopathological analysis of these candidate genes using data from TCGA." but you only showed their impact on survival, no association with other clinical features which you have for instance mentioned in ST2!

You should also include that type of analysis involving clinical data from TCGA database. By that you could additionally assess their "association with HNSCC molecular pathogenesis".

Line 139: To be honest, you must first perform a multivariate Cox regression analysis including all 5 six genes together to declare them as independent prognostic factors in patients with HNSCC! In your case they are independent from clinicopathological characteristics, but potentially not from each other!

Lines 141-144 and 169-172 are mere repetition!

Wherever you provide Spearman's corr. coef. (rho) provide also its P-value, because only statistically significant rho (P<0.05) should be commented.

Line 235: It is written "GEPIA2 database analyses showed..." while nowhere were those results presented, analysis described, GEPIA tool cited, etc.

Line 268: Its is written "The reagents used in this study are listed in Table S4." while in Table S4 there are listed features of fibroblast and HNSC cell lines!

Line 283: What does "...by factoring in clinical information from TCGA analyses." mean?

Line 288: From the text it is not clear for what the statistical language R and which packages were used?

Line 290: Put Spearman’s rank test where it belongs. Describe also how Kaplan-Meier plots were compered and how multivariate Cox regression analyses were performed.

Author Response

Please see

IJMS (ijms-1216243) revise letter

May 27, 2021

Dr. Nijiro Nohata

Guest Editor

Dear Dr. Nohata

We would like to express our gratitude for your consideration of our above-mentioned manuscript for publication in IJMS. Enclosed, please find the revised manuscript (ijms-1216243) along with a detailed explanation of the revisions, which were made based on the reviewers’ comments. All changes are highlighted in the revised manuscript.

Reviewer #1

In the submitted manuscript Oshima et al., by combining bioinformatical analyses of new and publicly available miRNA and mRNA expression data, together with in vitro assays, discovered that putative miR-31-5p and miR-31-3p target genes CACNB2, IL34, CGNL1, CNTN3 and GAS7 have potential tumor suppressive role in head and neck squamous cell carcinoma (HNSCC), and that inhibition of those two miRNAs decrease the proliferation, migration and invasion of HNSCC cells.

Although the presented results are somehow interesting, this study is quite simplistic and the hypothesis lacks more solid biological basis.

I have tree major concerns with this manuscript:

Comment-1: I still don't know why the authors have studied particularly those two miRNAs when they discovered 168 miRNAs up-regulated in HNSCC tissue samples, of which miR-31-5p/3p are somewhere in the middle of list according to logFC value! And what about significant down-regulated miRNAs and up-regulated mRNAs?! In the present form this manuscript looks more like a salami publication.

Response: The bitter comments from you are important to our study. There was a lack of explanation for the legitimacy of selecting miR-31-5p and miR-31-3p on this study. We continue to analyze microRNAs based on a clear research policy.

We are focusing on double strands of miRNAs (the guide strand and passenger strand) derived from pre-miRNAs. Based on the miRNA signature created this study, we selected the miRNAs with upregulated of both strands. Then, we targeted miRNA whose expression was confirmed to be upregulated by TCGA database analysis. The following text is added to the revised version to explain why we chose miR-31-5p and miR-31-3p in this study.

  1. Results

2.1. Identification of the miRNA expression signature of HNSCC by RNA sequencing Bulleted lists look like this:

Six cDNA libraries (derived from three HNSCC tissues and three normal oral epithelial tissues) were analyzed by RNA sequencing. After a trimming procedure, 955,347–1,927,436 reads were successfully mapped to the human miRNAs (Table S1). The clinical features of the specimens using in this study are summarized in Table S2.

A total of 168 miRNAs were identified as upregulated (log2 fold change > 1.5) in HNSCC tissues (Figure 1A and Table S3).

2.2. Expression levels and clinical significance of miR-31-5p and miR-31-3p in HNSCC

We focused on miRNAs of which both strands (the guide strand and passenger strand) derived from pre-miRNAs were upregulated in this signature. A total of 7 pre-miRNAs (miR-31, miR-223, miR-4655, miR-4781, miR-6753, miR-6830, and miR-6871) were detected in this signature (Table S3). From TCGA-HNSC database analysis, it was confirmed that miR-31 is only miRNA whose expression of both strands were significantly upregulated in HNSCC tissues among 7 pre-miRNAs (Figure 1B). The expression of neither miRNA was associated with worse overall survival rates in patients with HNSCC according to analysis of TCGA-HNSC data (Figure 1C).

In this study, we focused on miR-31-5p and miR-31-3p, and to investigate the functional analysis of these miRNAs.

Comment-2: Such large tables like Table 1, 2A and 2B are inappropriate to be placed in the main manuscript. It would be better to have heatmaps or PCA plots for distinguishing types of used samples based on the whole transcriptome. Those tables should be provided as supplementary. Also, since authors were interested only in down-regulated genes, there is an error in Table 2A and 2B column name: "(log2> 2.0)"!

Response: We would like to thank the reviewer's constructive comments. According to the reviewer’s suggestion, change Table-1 to S-Table-3. Then, upregulated miRNAs in this signature are presented as a heatmap in Figure 1A.

A list of target genes controlled by miRNA is important in miRNA research. It is informative to present the list of targets genes (Table 1A and 1B), and the tables will be presented as it is.

We appreciate your advice. We could correct Table 1A and 1B column name to "(log2< -2.0)"

Comment-3: Authors have potentially used to many self-citations, especially when describing methods (References 14-17), so now potential reader cannot completely and clearly follow what authors have done newly and what are their previous results! When someone looks at their published papers, it could be seen that they have previously used much more HNSCC tissue samples so it is not clear why now they used only 6?! In continuation with my previous comment, this manuscript shows elements of "cherry picking".

Response: We have published several articles of exploring the target genes of miRNAs in HNSCC. These articles showed the consistent methods of the experiments and we selected four new articles concerning with HNSCC. If someone read these articles, they could immediately understand how we experimented. No other articles are appropriate for the references in the material and method section.

In this study, we newly created RNA-sequencing based miRNA expression signature using oral squamous cell carcinoma (OSCC) specimens with no treatment. Currently, we are creating miRNA expression signature for each site of HNSCC to search for miRNAs that are characteristic of each site.

Although you pointed out about our sample quantity, it is adequate because we confirmed our expression signatures using numerous cohort data sets from TCGA.

Comments: To address some other bigger or smaller drawbacks:

Response: We would like to thank you for carefully pointing out our paper. The reviewer's point is very important.

Line 21: It is not evident how were those genes "closely associated with HNSCC molecular pathogenesis" when you only showed potential impact on an OVERALL survival (not even a disease-specific)?

Response: I would like to thank you for your pointing out.

As you pointed out, we didn’t verify the function of our target genes in HNSCC molecular pathogenesis. Currently, we focused on CGNL1 (miR-31-3p regulation), and we are in the process of establishing a stable cell lines. A detailed functional analysis of the target gene is planned for the next paper. We changed line 21 to the following text.

As suggested by the reviewer’s comment, we performed some clinicopathological analyzes using TCGA datasets. Regarding the disease-free survival rates, we have obtained positive data, and are shown in the Figure 5A and 5B.

Among these targets, the low expression of seven genes (miR-31-5p targets: CACNB2 and IL34; miR-31-3p targets: CGNL1, CNTN3, GAS7, HOPX, and PBX1) was closely associated with poor prognosis in HNSCC.

Line 31: Please provide the newest Cancer Statistics paper: https://pubmed.ncbi.nlm.nih.gov/33433946/

Response: We would like to thank the reviewer's suggestion in detail. But the newest statistics paper you provided shows the epidemiology of HNSCC in USA. It is more appropriate to show worldwide data.

Line 53 and throughout the text: Please do not use word "signature" when describe or present only 1-2 miRNA/mRNA expressions alone! It would be appropriate if you have shown that combinatorial expression of several miRNAs/mRNAs together have relation to HNSCC!

Response: We would like to thank you for your advice on our article. Your advice will be reflected in our future research. On the other hand, please be aware that our style also exists as we continue our research. For matters that have not been pointed out in the peer review so far, we will proceed with the current style. We are grateful for your understanding and cooperation.

Line 64: It is not clear why your miRNA expression data is not deposited to GEO database while gene (mRNA) expression is (GSE172120)?! Or this is just not clearly described in the text...

Response: Currently, we are researching on down-regulated miRNAs in HNSCC using the same data. We plan to register our miRNA expression data to GEO database when we submit the next treatise.

Line 67: Put "features of the HNSCC specimens".

Response: Thank you for your advice. Correct the text as instructed.

The clinical features of the HNSCC specimens using in this study are summarized in Table S2.

Line 68: Again it is not clear why you focused only on up-regulated miRNAs?!

Response: We have focused on miRNAs whose expression is down-regulated in cancer in our previous studies. In this time, one of the purposes is to establish a method for analyzing up-regulated miRNAs in cancer.

Line 77 and throughout the text: It is not clear which all or particular TCGA dataset(s) you have used?! Is it only TCGA-HNSC? Please provide a citation for this dataset: https://www.nature.com/articles/nature14129

Also clearly state on figures which TCGA dataset you used for the analysis presented on the figure, not only "HNSCC" because it could be ambiguous.

Response: The TCGA database handles a larger number of cases(n=485) than the dataset(n=279) you presented, so it makes more sense to use the HNSCC set of TCGA database. Details of the dataset used are mentioned in Material and method 4.7 as described below. Then add ‘HNSC’ to the figures created using the TCGA database.

For the Kaplan–Meier survival analysis, we downloaded TCGA-HNSC clinical data (TCGA, Firehose Legacy) from cBioportal (https://www.cbioportal.org) on April 10, 2020. Gene expression data for each gene were collected from OncoLnc (http://www.oncolnc.org). R ver. 4.0.2 (R Foundation for Statistical Computing, Vienna, Austria) was used for the statistical analyses.

Line 94: Describe in the text why you chose particularly SAS and HSC-2 HNSCC cell lines because when you look at Figure S1 you can see that they do not have the highest mean expression levels of miR-31-5p and miR-31-3p!

Such large dispersion (standard deviation) for miRNA expresion in a single cell line is strange and unacceptable!

Also, where are the supplementary figures' legends?

Response: I would like to thank you for your pointing out.

As you pointed out, other cell lines, for example Ca9-22 and  HSC-4, showed higher expression levels of miR-31-5p and miR-31-3p than SAS and HSC-2. The difference of expression levels in those cell lines is not important because it is clear that those cell lines showed much higher expression compared to normal fibroblasts.

We chose SAS and HSC-2 in HNSCC cell lines because two cell lines proliferate well for functional assay.

As you indicated, large dispersion is shown in supplementary figure S1 in some cell lines. We suspect that those measurement variabilities caused because of our experimentally procedure. If we performed qRT-PCR several times, we would be able to make the dispersion smaller. But it is not essential because the figure  showed clearly that expression levels in HNSCC cell lines  were higher than in normal fibroblast cell lines.

We added supplementary figures' legends to the revised version.

Line 96: Although you mostly cite your previous papers when describing methods, some more information are needed in general.

What is "mock" and what "control"?

Regarding the Supplementary Table 3, for what were those Pre-miR miRNA Precursors used?

Response: We mean "mock" is a cell line that does not transfect precursor, inhibitor and uses only reagents, and "control" refers to a cell line that has been transfected with a negative control precursor.

The negative precursor we used for the "control" is adding to the supplemental table 4.

In supplemental table 4, we mentioned the unused precursor at the submit time. We sincerely apologize. But since it is used in Fig. 8 assays added this time, so we’d like to leave it as described.

Lines 97-98: Something is missing in that sentence.

Response: We would like to thank you for carefully pointing out our paper. We added the following sentence in lines 97-98.

To evaluate their effects in functional assays, we confirmed the expression of miR-31-5p and miR-31-3p after transfection of inhibitors into SAS and HSC-2 cells (Figure S2).

Line 119: Why experimentally proven data about miRNA:mRNA interactions were not used? For instance, data from TarBase v.8 database would provide more plausibility and importance of your selection of genes.

Response: We would like to thank the reviewer's constructive comments. We tried TarBase v.8 database, and applied hsa-miR-31-5p/3p to search related genes. Filtering by terms of Head/Neck in tissue, no results found. While TarBase database shows interactions between miRNA and mRNA in previous studies, our aim is to find out newly molecular pathogenesis of HNSCC. So we used expression data sets from HNSCC clinical tissue and screened miR-31-5p/3p targets as shown in Figure 3.

Line 123: You said that you have "performed a clinicopathological analysis of these candidate genes using data from TCGA." but you only showed their impact on survival, no association with other clinical features which you have for instance mentioned in ST2!

You should also include that type of analysis involving clinical data from TCGA database. By that you could additionally assess their "association with HNSCC molecular pathogenesis".

Response: As suggested by the reviewer’s comment, we performed some clinicopathological analyzes using TCGA datasets. Regarding the disease-free survival rates, we have obtained positive data, and are shown in the Figure 5.

Line 139: To be honest, you must first perform a multivariate Cox regression analysis including all 5 six genes together to declare them as independent prognostic factors in patients with HNSCC! In your case they are independent from clinicopathological characteristics, but potentially not from each other!

Response: I think your point is very important issue in our study. There are various decisions about when to perform multivariate analysis. Following your suggestions, I analyzed the genes that have significantly in 5-year survival rate. Multivariate cox regression analyses revealed that expression of 2 genes (CACNB2 and CGNL1) were independent prognostic factors for patients with HNSCC. However, this paper presents the results of multivariate analysis for each of the genes involved in the 5-year survival rate as same as our previous studies. We will continue to work with statisticians on multivariate analysis. We are grateful for your understanding and cooperation.

Lines 141-144 and 169-172 are mere repetition!

Wherever you provide Spearman's corr. coef. (rho) provide also its P-value, because only statistically significant rho (P<0.05) should be commented.

Response: We delete the description in the text and add the P-value and Spearman's corr. coef. (rho) to the Figure7 legend.

Figure7 legend.

Spearman’s rank test indicated negative correlations of miR-31-5p expression with their targets (CACNB2/miR-31-5p: p < 0.001, r = -0.3748; IL34/miR-31-5p: p < 0.001, r = -0.5296). Similarly, negative correlations were detected in miR-31-3p expression with their targets (CGNL1/miR-31-3p: p < 0.001, r = -0.5145; CNTN3/miR-31-3p: p < 0.001, r = -0.3601; GAS7/miR-31-3p: p < 0.001, r = -0.3170)

Line 235: It is written "GEPIA2 database analyses showed..." while nowhere were those results presented, analysis described, GEPIA tool cited, etc.

Response: I would like to thank you for your pointing out.

Few previous studies revealed the role of CGNL1 in cancer, we used GEPIA2 database.

We change line 235 to the following text.

GEPIA2 database ( http://gepia2.cancer-pku.cn/#index ) analyses showed that expression of CGNL1 was significantly downregulated in cervical squamous cell carcinoma, esophageal carcinoma, and lung squamous cell carcinoma, suggesting that CGNL1 is downregulated in human squamous cell carcinoma.

Line 268: Its is written "The reagents used in this study are listed in Table S4." while in Table S4 there are listed features of fibroblast and HNSC cell lines!

Response: Thank you for your advice. As we changed figures and tables for revised paper, the reagents used in this study are listed in Table S4 finally.

Line 283: What does "...by factoring in clinical information from TCGA analyses." mean?

Response: As explained in the text and Figure 3, it refers to the 5-year survival rate and multivariate analysis.

Line 288: From the text it is not clear for what the statistical language R and which packages were used?

Line 290: Put Spearman’s rank test where it belongs. Describe also how Kaplan-Meier plots were compered and how multivariate Cox regression analyses were performed.

Response: I would like to thank you for your pointing out. We added the information of the language R and packages. We also merged 4.6. and 4.7. as “Analysis of the clinical significance of HNSCC patients using TCGA-HNSC data” and wrote statistical analysis in detail as following text.

4.6. Analysis of the clinical significance of HNSCC patients using TCGA data

The strategy used to identify miRNA target genes is presented in Figure 3. We selected putative target genes with miR-31-5p and miR-31-3p binding sites using TargetScanHuman ver. 7.2 (http://www.targetscan.org/vert_72/; data were downloaded on 10 July 2020). The expression profiles of HNSCC clinical specimens (genes downregulated in HNSCC tissues) were used for screening miRNA target genes. Our expression data were deposited in the GEO database (accession number: GSE172120). Furthermore, we narrowed down the candidate genes by factoring in clinical information from TCGA analyses.

For the Kaplan–Meier survival analysis, we downloaded TCGA clinical data (TCGA, Firehose Legacy) from cBioportal (https://www.cbioportal.org) on April 10, 2020. Gene expression data for each gene were collected from OncoLnc (http://www.oncolnc.org). For the log-rank test, we used R ver. 4.0.2 (R Foundation for Statistical Computing, Vienna, Austria) , and “survival” and “survminer” packages.

Multivariate Cox regression analyses were also performed using TCGA-HNSC clinical data and survival data according to the expression level of each gene from OncoLnc to identify factors associated with HNSCC patient survival. In addition to gene expression, the tumor stage, pathological grade, and age were evaluated as potential independent prognostic factors. The multivariate analyses were performed using JMP Pro 15.0.0 (SAS Institute Inc., Cary, NC, USA).

4.7. Statistical analysis

Statistical analyses were performed using GraphPad Prism 7 (GraphPad Software, La Jolla, CA, USA) and JMP Pro 15 (SAS Institute Inc., Cary, NC, USA). Dunnet’s test were used for multiple group comparisons. For correlation analyses, Spearman’s test was applied.

Thank you for your constructive comments and suggestions.

We believe that our manuscript has been greatly improved and is now suitable for publication in IJMS. Again, thank you for your consideration of our manuscript for publication in your journal.

Sincerely yours,

Naohiko Seki, Ph.D.

Department of Functional Genomics

Chiba University Graduate School of Medicine

1-8-1 Inohana, Chuo-ku,

Chiba 260-8670, Japan

Phone: +81-43-226-2971

Fax: +81-43-227-3442

the attachment.

Reviewer 2 Report

In this study, Oshima and colleagues found that miR-31-5p and miR-31-3p play important roles in the oncogenesis of head and neck squamous cell carcinoma. The authors claimed that the inhibition of miR-31-5p and miR-31-3p attenuated cancer cell malignant phenotypes; however, I do not agree with this claim based on the presented data.

In Fig. 1B, the authors showed the expression levels of miR-31-5p and miR-31-3p were not associated with overall survival rates in patients with HNSCC at all.

In Fig. 2, the authors showed that the inhibition of miR-31-5p and miR-31-3p attenuated cancer cell proliferation, migration, and invasion; however, these changes were very mild. These data are not convincing.

Author Response

IJMS (ijms-1216243) revise letter

May 27, 2021

Dr. Nijiro Nohata

Guest Editor

Dear Dr. Nohata

We would like to express our gratitude for your consideration of our above-mentioned manuscript for publication in IJMS. Enclosed, please find the revised manuscript (ijms-1216243) along with a detailed explanation of the revisions, which were made based on the reviewers’ comments. All changes are highlighted in the revised manuscript.

Reviewer #2

In this study, Oshima and colleagues found that miR-31-5p and miR-31-3p play important roles in the oncogenesis of head and neck squamous cell carcinoma. The authors claimed that the inhibition of miR-31-5p and miR-31-3p attenuated cancer cell malignant phenotypes; however, I do not agree with this claim based on the presented data.

Comment-1: In Fig. 1B, the authors showed the expression levels of miR-31-5p and miR-31-3p were not associated with overall survival rates in patients with HNSCC at all.

Response: The reviewer's point is accurate. Unfortunately, miR-31-5p and miR-31-3p expressions do not affect the prognosis of HNSCC patients. However, the target molecules controlled by these miRNAs affect the prognosis of HNSCC patients. In addition, in this paper, miRNA is selected based on the fact that both strands (guide strand and passenger strand) are upregulated. The miRNAs that met this criterion were miR-31-5p and miR-31-3p in this signature.

The following text and table are added to the revised version to explain why we chose miR-31-5p and miR-31-3p in this study.

  1. Results

2.1. Identification of the miRNA expression signature of HNSCC by RNA sequencing Bulleted lists look like this:

Six cDNA libraries (derived from three HNSCC tissues and three normal oral epithelial tissues) were analyzed by RNA sequencing. After a trimming procedure, 955,347–1,927,436 reads were successfully mapped to the human miRNAs (Table S1). The clinical features of the specimens using in this study are summarized in Table S2.

A total of 168 miRNAs were identified as upregulated (log2 fold change > 1.5) in HNSCC tissues (Figure 1A and Table S3).

2.2. Expression levels and clinical significance of miR-31-5p and miR-31-3p in HNSCC

We focused on miRNAs of which both strands (the guide strand and passenger strand) derived from pre-miRNAs were upregulated in this signature. A total of 7 pre-miRNAs (miR-31, miR-223, miR-4655, miR-4781, miR-6753, miR-6830, and miR-6871) were detected in this signature (Table S3). From TCGA-HNSC database analysis, it was confirmed that miR-31 is only miRNA whose expression of both strands were significantly upregulated in HNSCC tissues among 7 pre-miRNAs (Figure 1B). The expression of neither miRNA was associated with worse overall survival rates in patients with HNSCC according to analysis of TCGA-HNSC data (Figure 1C).

In this study, we focused on miR-31-5p and miR-31-3p, and to investigate the functional analysis of these miRNAs.

Comment-2: In Fig. 2, the authors showed that the inhibition of miR-31-5p and miR-31-3p attenuated cancer cell proliferation, migration, and invasion; however, these changes were very mild. These data are not convincing.

Response: Thank you for your comments, but I will refute your point. In this analysis, we judge that the inhibitors (miRNA-31-5p and miR-31-3p antisense) had decent cancer-suppressing effects. In particular, miR-31-3p inhibitor significantly suppressed the migration and invasion abilities of cancer cells. The tumor suppressor effect of this paper is consistent with the results of our previous miRNA analyses.

Thank you for your constructive comments and suggestions.

We believe that our manuscript has been greatly improved and is now suitable for publication in IJMS. Again, thank you for your consideration of our manuscript for publication in your journal.

Sincerely yours,

Naohiko Seki, Ph.D.

Department of Functional Genomics

Chiba University Graduate School of Medicine

1-8-1 Inohana, Chuo-ku,

Chiba 260-8670, Japan

Phone: +81-43-226-2971

Fax: +81-43-227-3442

he attachment.

Round 2

Reviewer 1 Report

The authors have substantially improved the quality of their manuscript but few issues still have to be improved:

1) I have suggested that reference 2 should be replaced with the newest version: Cancer statistics, 2021 instead of 2019. They are both about "the American Cancer Society estimates the numbers of new cancer cases and deaths that will occur in the United States" and not "worldwide data".

2) Table S5 (line 125) is mentioned before Table S4 (line 134). Re-check that all figures and tables are properly numerated and consecutively cited in the text.

3) Line 246 and Figure 8C: Syntagm "deletion vector" is kind of awkward. Rather use description like "vector with partially deleted CGNL1 3′-UTR" and "Deleted-type".

4) State somewhere in the text what is presented on bar graphs. I suppose mean value + standard deviation.

5) State in 4.8 subsection which P-values were considered statistically significant.

6) State in the legend of Fig S3 what those asterisks mean.

7) I again kindly ask authors to properly acknowledge creators of all data and databases they have used in their study by providing references for TCGA-HNSC data, GEPIA2, TargetScan, etc.

Reviewer 2 Report

I don't have any more questions. I recommend that the supplementary figures S2-S4 can be incorporated into the main figures.
